# Trusted Data Storage Architecture for National Infrastructure

**DOI:** 10.3390/s22062318

**Published:** 2022-03-17

**Authors:** Yichuan Wang, Rui Fan, Xiaolong Liang, Pengge Li, Xinhong Hei

**Affiliations:** 1School of Computer Science and Engineering, Xi’an University of Technology, Xi’an 710048, China; chuan@xaut.edu.cn (Y.W.); 2211221145@stu.xaut.edu.cn (R.F.); 2191221082@stu.xaut.edu.cn (X.L.); 2190321308@stu.xaut.edu.cn (P.L.); 2Shaanxi Key Laboratory for Network Computing and Security Technology, Xi’an 710048, China

**Keywords:** blockchain, federated learning, knowledge extraction, water conservancy project

## Abstract

National infrastructure is a material engineering facility that provides public services for social production and residents’ lives, and a large-scale complex device or system is used to ensure normal social and economic activities. Due to the problems of difficult data collection, long project period, complex data, poor security, difficult traceability and data intercommunication, the archives management of most national infrastructure is still in the pre-information era. To solve these problems, this paper proposes a trusted data storage architecture for national infrastructure based on blockchain. This consists of real-time collection of national infrastructure construction data through sensors and other Internet of Things devices, conversion of heterogeneous data source data into a unified format according to specific business flows, and timely storage of data in the blockchain to ensure data security and persistence. Knowledge extraction of data stored in the chain and the data of multiple regions or fields are jointly modeled through federal learning. The parameters and results are stored in the chain, and the information of each node is shared to solve the problem of data intercommunication.

## 1. Introduction

The national infrastructure construction [1] involves a large number of key data, for example, a large number of engineering archives are generated during the construction of water conservancy projects. These documents are important basic information for project quality control and problem tracing, and are related to the safety of the whole life cycle of the project. At present, the status of basic construction archives management is almost the same as that of the pre-information era, and many problems need to be solved urgently.

There is a huge amount of paper. The volume of construction archives of water conservancy projects is extremely huge. The earth and stone volume of a medium-sized reservoir dam can reach several million cubic meters and the grouting volume can reach more than 100,000 m. At present, the related construction archives are still mainly in paper form, and there is a clear gap compared with the industries such as finance, medical care, transportation and logistics. Thousands of handwritten construction documents are piled up in the archives. Although the state has issued relevant legal documents on electronic signature and electronic filing of documents, the digitalization of construction files and e-government management in the field of capital construction are still struggling.There is an issue of lax management and lack of credibility. Incomplete files occur frequently. It is common that data are not filled in and stored according to the specification. Some information may be falsified.The is poor traceability. The cataloging and archiving of engineering construction archives has not yet formed a set of strict and complete disciplines and management systems. Once there is a need for traceability, it is often impossible to achieve. On the afternoon of 5 June 2010, a huge landslide occurred in Jiwei Mountain, an iron mine township in Wulong County, Chongqing, and the hole was buried. Engineers transferred a powerful drilling rig, hoping to deliver food to the people in the cave through drilling holes, but the rescue work was not successfully implemented because the relevant drawings of the tunnel location could not be found. The suspected bean curd residue project revealed after the Wenchuan earthquake could not find the construction data of that year and could not carry out an in-depth investigation.The security is worrying. Most of the construction files do not have strict security control measures, and there are hidden dangers of man-made attack and destruction.

In addition, take the train operation control system as an example. Once safety problems occur in the train operation control system, the safety guarantee of high-speed train operation, the consequences are unimaginable. In 2004, the Australian Railway Company’s train operation control system was maliciously attacked, which made it impossible for the driver to communicate with the signal box, causing 300,000 passengers to be trapped. In 2008, a Polish citizen used a jammer to interfere with the rail trains in Loz, which derailed five trains and caused a lot of property losses. These cases show that the safety of train operation control systems is of great significance to the efficient operation of trains.

The lag of informatization management of national infrastructure archives is determined by its particularity. The management of infrastructure archives is closely related to the engineering quality and the safety of people’s lives and property, and has a high degree of anti-tampering and anti-repudiation requirements. Although the electronic signature is guaranteed by the state from the legal level, the digital (non-paper) construction technical documents are transmitted on the Internet and stored on the centralized server for a long time, which inevitably leads to serious trust and security concerns. The higher the sensitivity of data, the greater the risk of “hacking” it with high-tech means. Blockchain technology uses digital encryption and block structure to store data. On the one hand, it can realize the electronic storage of paper materials; on the other hand, distributed storage can be used to avoid data tampering, and block information can be used to realize data traceability and solve the management problem of infrastructure data.

This paper proposes a trusted data storage architecture for national infrastructure based on blockchain. Real-time data collection by sensors and other IoT devices [2], as well as knowledge extraction and federal learning are used to analyze the data, and blockchain technology is used to build a credible execution environment to ensure the security and traceability of national infrastructure data. The specific contributions of the scheme are as follows:

Firstly, real-time collection of national infrastructure data ensures the accuracy and credibility of the data source. A large number of heterogeneous data are uniformly formatted to meet the data specification of blockchain. Data are stored in the blockchain, which ensures the safe storage and traceability of data. Secondly, the technologies such as knowledge extraction and federal learning are combined with blockchain technology to ensure the effective use of data and credibility of federated learning. Finally, the smart contract module in the blockchain module is extended, and a calling module is added to realize the interaction with knowledge extraction, federal learning and other modules with high flexibility. In addition, this architecture has been applied to train operation control systems and water conservancy projects, and good results have been achieved.

## 2. Materials and Methods

### 2.1. Blockchain

Satoshi Nakamoto [3], the founder of Bitcoin, first put forward the concept of digital currency in 2008. After that, the blockchain, as the technical support of digital currency, began to appear and be applied in the financial industry, Internet of Things logistics, public services, education, network security [4] and other industries. Blockchain is widely used in various industries due to the following advantage. The first is decentralization, i.e., the multi-node maintenance of the network, where each user node stores a ledger separately to prevent data loss caused by single node downtime. The second is that it is tamper-proof. Blockchain is a chain structure in which each block formed by block linking is connected to the blocks before and after it in chronological order. All the uplink data are based on the digital signature technology in cryptography, and the uplink data cannot be tampered by a third party and cannot be denied. If the information data are modified or deleted, the consent of more than 51% of the nodes in the whole network is required, but it is difficult to control more than 51% of the nodes in the whole network, thus ensuring the security of blockchain data to a certain extent. The third is that it is independent. Smart contracts ensure the independence and credibility of blockchain. A smart contract is an automatically executable programming code rule [5], which will not change when added to every node on the network for operation, and no third-party trust mechanism is needed. Rules in smart contracts are automatically executed, the operation is open, and the subject can be supervised and traced. Smart contracts can meet the needs of different services, and can be adjusted according to different business scenarios, which are flexible. Therefore, such as Chen [6] and Manimaran [7], many scholars combine blockchain with the financial field to ensure the security and transparency of scheme data. Bai [8] combines blockchain and edge computing [9,10,11], which makes the access control of the Internet of Things more reliable and secure. Wang et al. [12] applied blockchain technology to the networking technology of mobile vehicles to ensure the security and privacy of the vehicle network. Shahbazi [13] studied vehicle safety and demand service, and used the combination of data analysis and blockchain to prevent false transactions of dishonest users.

### 2.2. Knowledge Extraction

Knowledge extraction is to create knowledge from structured and unstructured sources; that is to say, the useful knowledge contained in information sources is extracted through the processes of recognition, understanding, screening and induction, and stored to form a knowledge element library. Knowledge extraction can provide material for knowledge modeling. These materials are not all “raw”, and a kind of structured data, such as marks, charts, glossary, formulas and informal rules, can be obtained by using extraction technology.

This information extraction process converts unstructured information embedded in text into structured data, for example, for filling relational databases to support further processing. The process of knowledge extraction is mainly divided into two parts. The first part is the recognition of knowledge entities, named entity recognition [14] (NER). NER is to find each named entity mentioned in the text and mark its type. Once all named entities in the text are extracted, they can be linked to the set corresponding to the actual entities, from early dictionary and rule-based methods to statistical machine learning methods. The methods based on statistical machine learning mainly include Hidden Markov Model (HMM), Maximum Entropy Model (MEM), Support Vector Machine (SVM), Conditional Random Field (CRF) and so on [15,16,17,18]. Conditional field [19] is the mainstream model of NER at present. The second part is relationship extraction: relationship extraction is to extract the semantic relationship between two or more entities from the text. Wang et al. [20] proposed a relational attention model to extract the relationships between sentences. Although machine learning (neural or MEM/CRF) sequence model is the norm of academic research, NER’s business methods are usually based on the practical combination of lists and rules.

### 2.3. Federal Learning

Federal learning is the local training and analysis of common data, and the selective display of training results, on the premise that multiple participants guarantee that their private data will not be leaked [21,22,23,24,25]. Federal learning can model data usage and machine learning under the requirements of user privacy protection, data security and government, which can effectively solve the problem of data islands. According to different modes, federal learning can be divided into horizontal federal learning, vertical federal learning and transfer federal learning [26].

At present, there is also a lot of research work on the combination of federal learning and blockchain. Martinez et al. [27] proposed to use EOS blockchain as the incentive layer of federal learning to encourage more people to invest enthusiasm and high-quality data for maintenance. Lu et al. [28] proposed a secure data sharing architecture based on authorization of blockchain car networking, which protects the privacy of shared data and improves the utilization rate of system computer resources through feedback learning. Yin et al. [29] applied blockchain and federated learning to the Internet of Things [30] environment to implement a secure data cooperation framework, separate public data from private data and protect the safe use and transmission of data in the blockchain.

As can be seen from Table 1, many scholars combine blockchain technology with federal learning to ensure data security. However, it can be seen that no one has designed a national infrastructure data architecture at present, because of the complexity of national infrastructure data collection and processing, long construction period, poor security, difficulty in traceability and low degree of informatization. Therefore, this paper designs a trusted storage architecture for national infrastructure data, adopts an NER method based on list and rules, and adds a little supervised machine learning technology to extract knowledge. Combining blockchain technology with federated learning, we build a credible federated learning execution environment, save the results of knowledge extraction and data training, and ensure the credibility and traceability of data.

## 3. Scheme and Process

As shown in Figure 1, the trusted storage architecture for national infrastructure data is mainly divided into data collection, business flow analysis, trusted storage and data application modules. First, the underlying data are collected by sensors, paper files, audio and video, and then business process analysis is carried out. Second, according to the characteristics of business process, on the one hand, a traditional database of blockchain is used for data persistence, on the other hand, knowledge extraction and federated learning are carried out on blockchain. In the application layer, traditional business interface operations such as forensic tracking and feature recognition can be performed. The modules of the trusted storage part are described as follows.

The blockchain module is a blockchain technology based on open source Hyperledger Fabric as the bottom layer. Among them, the chaincode is divided into service chaincode, system chaincode and control chaincode. Service chaincode is responsible for implementing specific business rules. System chaincode is responsible for node endorsement and transaction, block query and code life cycle. Control chaincode interacts with other module codes by calling the framework gRPC. In this scheme, two business chain codes are designed, one is responsible for the storage of primary data, and the other is responsible for the storage of training data after extraction. Different role operation rules are implemented by different business chaincodes. Through the blockchain, it can adapt to different scenarios with strong adaptability. Blockchain module ensures the trusted storage and tamper-proof nature of federal learning data, training parameters and models, and provides traceability services for malicious behaviors and fault operations in the training process, creating a good and safe environment for federal learning and ensuring the quality of training models.

The federal learning module is a new artificial intelligence infrastructure, which can ensure the security of data interaction and terminal privacy data, and ensure the legitimacy of multi-node participants’ efficient machine learning. The training mode of federal learning can be divided into two ways: client training and on-chain training. The client carries out chain training according to the user’s needs, stores its training parameters and models locally, and stores the parameter information and models that need to be shared in the blockchain for other users to use. On-chain training is carried out in the central node, which has strong data processing and storage capabilities, and stores the parameters and models needed for the whole network training. Complex training is carried out on the chain. Blockchain ensures the security and tamper-proof ability of models and parameters and ensures the data security and credibility of federated learning.

The information extraction module interacts with the business process analysis module. The business process analysis module generates data with a uniform format, extracts knowledge from these data, and generates reliable data that can be used by a given program. These data can be used as a resource and can be used by the federal learning module. The data extraction module uses the NER method based on lists and rules, adds a little supervised machine learning, and uses the frequent itemset mining algorithm. The frequent itemset mining algorithm finds redundant data items in a standardized format by judging the periodicity and density of the data according to the redundant data, and preliminarily marking the data. The supervised machine learning training algorithm trains labeled data through three classifiers and filters redundant data.

## 4. Detailed Description of the Scheme

### 4.1. Business Flow Generation Blockchain

The description based on business flow is expressed in the form of an xml file. After reading and sorting out some documents, we know that BML [31] (a data mapping language) proposed by Ba-Lam Do et al. is data conversion mapping from a traditional data storage mechanism to a blockchain network, which can accept xml input data sources and support two output platforms, including Ethereum and Hyperledger Fabric.

BML aims to be a universal mapping language, which allows heterogeneous data sources to be transformed into the blockchain by user-specified rules, and then queries such data from the blockchain network. Therefore, we designed BML with six remarkable features: uniformity, efficiency, convenience, inheritance, openness, and relationship.

Figure 2 shows the architecture of the BML system. The architecture is divided into user interface module, engine module and blockchain module. The user interface module is responsible for the user’s interaction with the program, sending the user-defined descriptive files to the engine module according to the established syntax or putting the descriptive files into the API request body for data conversion, and returning the data to the user after the data conversion is completed. The engine module in BML language is a central processing layer composed of a group of processors, adapters and databases. The processing is related to the number of CPU cores. In parallel operation, data are collected according to the mapping definition of user life, and the data are divided. If the number of CPUs is n, divide the data into n copies. Then, the data are converted into a common format and transmitted to the blockchain module according to the advanced encryption standard (AES). When reading, you can query data through CID, then locate and search, decode and filter the found data in parallel, and return the filtered results to users. Blockchain module consists of nodes and chaincode. Through the node, you can interact with the blockchain and access the blockchain network for reading, writing, searching and other operations.

### 4.2. Federal Learning

The federal learning module is divided into two sub-modules, horizontal federal learning module and vertical learning module. Different federal learning methods are adopted according to different characteristics of national infrastructure data. For example, when the infrastructure mode is the same, but the participants are different, with the characteristics of more overlapping data features and less overlapping sample ID, the horizontal federal learning module is selected. On the contrary, when there are many overlapping sample IDs, but few overlapping digital features, different infrastructure modes and the same participants, the vertical federal learning module is selected.

Horizontal federal learning [32] is also called sample-based federal learning. Let the matrix $D_{i}$ represent the information of the i data owner, each row represents a sample, each column represents a feature, and some data sets are required to contain data label columns. x represents the feature space, y represents the range of label values, and J represents the ID space of samples. x, y and J together constitute a complete training data set. This can be described as: (1)$$x_{i}=x_{j},y_{i}=y_{j},J_{i}\neq J_{j},\forall D_{i},D_{j},i\neq j\;$$

In the horizontal federated learning system, *k* participants have the same data structure, and learn a machine learning model together through a parameter server or cloud server. Assuming that the participants are honest and the server is honest, the leakage from any participant to the server is not allowed. The training process of horizontal federal learning system includes the following four processes:Participants calculate the training gradient locally, use key sharing technology to mask the selected gradient, and then upload the masked result to the server.The server performs secure data aggregation without learning the information of any participant.The server returns the aggregation result to the participants.Participants use the decoded gradient to update their models.

Iterate these four steps until the loss function converges, and the whole training process is completed.

The federal average algorithm is suitable for any of the following finite sum loss functions:(2)$$\underset{w\in R^{d}}{min}f(w)=\frac{1}{n}\overset{n}{\underset{i=1}{\sum }}f_{i}(w)$$

n represents the number of training data, and w∈$R^{d}$ represents the model parameters of d dimension. $f_{i}\left(w\right)=\ell \left(x_{i},y_{i};\;w\right)$ is the predicted loss function and $\left(x_{i},y_{i}\right)$ is sample input.

In a horizontal federated learning system, it is assumed that data are distributed in *K* clients, where $n_{k}$ represents the data number set of client *k* and $P_{k}$ represents the data point set of client *k*. If $n_{k}=\left|P_{k}\right|$, then the data number set of client *k* is equal to the data point set, and Formula (2) can be rewritten as:(3)$$f(w)=\overset{K}{\underset{k=1}{\sum }}\frac{n_{k}}{n}F_{k}(w),F_{k}(w)=\frac{1}{n_{k}}\underset{i\in P_{k}}{\sum }f_{i}(w)$$

The above algorithm will expose the plaintext content of the intermediate result. It does not provide any security protection. If the data structure is also leaked, the leakage of model gradient or model parameters may lead to the leakage of important data and model information. Therefore, the federal learning module is put into the blockchain, and the data are encrypted and stored to ensure the safety and reliability of the results. 

### 4.3. Knowledge Extraction

Named entity recognition is basically based on the pure statistical sequence model in academic circles, but the method of processing named entity recognition in industry needs to be more practical. Supervised learning and some rules are added. The most commonly used method is to input the results of the previous sequence into the next sequence through the sequence.

The first step is to mark ambiguous named entities with high-precision rules.

The second step is to find the substring of the name found before.

The third step is to compare the list of words in a specific domain with the previously identified named entities.

Fourth, the probabilistic sequence labeling model is applied to take the previous labels as features.

The frequent itemset mining algorithm (Apriori algorithm) [33,34,35,36] used for I = {$i_{1}$, $i_{2}$,…, $i_{d}$} is the set of all items in the data, while T = {$t_{1}$, $t_{2}$,…, $t_{d}$} is the set of all transactions. A collection of 0 or more items is called an itemset. If a project set contains *k* projects, it is called *k* project set. Obviously, every transaction contains a subset of I. Association rules are implication expressions in the form of *X*→*Y*, where *X* and *Y* are disjointed itemsets, that is, *X* ∩ *Y* = ∅. The strength of association rules can be measured by its support and confidence. The minimum support is an artificially defined threshold, which indicates the lowest importance of itemsets in the statistical sense. The minimum confidence is also an artificially defined threshold, indicating the lowest reliability of association rules. Only when the support and confidence both reach the minimum support and the minimum confidence can this association rule be called a strong rule. All itemsets that meet the minimum support are called frequent itemsets. (All non-empty subsets of frequent itemsets are also frequent itemsets; if the A itemset is not a frequent itemset, then the union of other itemsets or transactions with the A itemset is not a frequent itemset). If all frequent itemsets are solved directly, the time complexity will be very high.

Support: determines how often rules can be used for a given data set.
(4)$$Support\left(X,Y\right)=P\left(X,Y\right)=\frac{num\left(xy\right)}{\;num(allsamples)\;}$$

Confidence: determines the frequency of *Y* in transactions involving *X*.
(5)$$Confidence\;\left(XY\right)=P\left(x|Y\right)=\frac{P\left(xy\right)}{P\left(y\right)}$$

Supervised machine learning repeatedly samples labeled sample sets to generate N labeled training sets, and generates N corresponding classifiers from the N training sets. In the cooperative training process, the newly labeled samples from each classifier are provided by other N − 1 classifiers. If N − 1 classifiers predict the same unlabeled sample, this sample has higher confidence, and it is added to the labeled training set of the remaining classifiers. Through the built-in optimization of N classifiers, the unlabeled instances are jointly predicted.

The process of supervised learning to realize relationship extraction is as follows: first, specific relationships and named entities are selected, and the training corpus manually labels relationships and named entities. Finally, the annotated corpus is used to train classifiers to label training sets that have not been seen before. The most direct method has three steps:

The first step is to find a pair of named entities in a sentence.

In the second step, the function of binary classifier is to judge whether there is a relationship between two named entities.

In the third step, the classifier will be used to mark the relationship between named entities.

Pseudo code is shown in Algorithm 1.
**Algorithm 1** FINDRELATIONS**Input:** words**Output:** relationsRelationa←nilentities←FINDENTITIES(words)forall entity pairs <ℯ1, ℯ2> in entities do  if RELATED? (ℯ1, ℯ2)    relations←relations + CLASSIFYRELATION (ℯ1, ℯ2)

## 5. Experimental Analysis

In this section, we take the water conservancy industry and a train control system in China as examples to show the application of trusted data storage architecture based on blockchain. The equipment and environment configuration used in this experiment are shown in Table 2.

Table 3 qualitatively compares the traditional storage architecture with the trusted storage architecture proposed in this paper. This architecture stores data in the blockchain to ensure data security, and builds a trusted execution environment to analyze data. It has certain advantages to ensure the security, traceability, intercommunication and full utilization of national infrastructure data. Traditional storage architecture generally uses centralized database to store data, which can store data completely. But once attacked, the data is easily leaked. The trusted storage architecture uses distributed storage of data, multi-node backup, and data is not easy to lose. Combined with blockchain technology, it is impossible to tamper with data and protect data.

### 5.1. Examples of Rail Transit Projects

Chinese Train Control System 3 (CTCS 3) [37] is the core component of China’s railway technical system and equipment. Efficient information exchange provides a powerful guarantee for the safe, reliable and efficient operation of high-speed trains. Figure 3 is the architecture of CTCS 3. The train operation control system mainly includes ground equipment (wireless block center, train control center, ZPW-2000 system track circuit and transponder) and on-board equipment (GSM-R wireless communication unit, track circuit information receiving unit and recording unit). During each train operation, real-time information will be exchanged between different devices in the train operation control system. Although the efficiency of wireless communication is very high, it will also bring a lot of network information security problems. The traditional Web-based key management method can usually verify data, but it cannot cope with the sensitive problem of single-point failure caused by centralization. Therefore, using the unique decentralized framework of blockchain, various distributed authentication systems are realized and used to counter the security risks of a centralized structure. Through the identity authentication mechanism and related key encryption scheme, the data in transmission are encoded and safely transmitted to ensure the integrity and non-repudiation of communication data. In addition, using trusted data storage architecture to store data safely can also extract useful knowledge from it for model training. This can fully guarantee the data security and make full use of the data.

The process of the trusted data storage architecture in the specific application of the train control system is as follows: First, obtain the safety log, data packet, sensor, flow data and other information generated in the train control system. Second, analyze the business process of the obtained data. Then, upload the data to the Fabric blockchain for persistent storage of the database. Furthermore, knowledge extraction and federal learning are carried out on the data, and the obtained results are stored in the blockchain. In addition, some changes have been made in the form of added functions such as API docking, analysis configuration customization and development customization. In addition, it has the functions of security situation detection and early warning, real-time visual display.

Figure 4 analyzes the traffic flow of the train control system. The wireless block center generates a driving license according to the track circuit and other messages. The driving license, line parameters and temporary speed limit are transmitted to CTCS 3 vehicle-mounted equipment through a GSM-R wireless communication system; at the same time, through the GSM-R wireless [38] communication system, messages such as location and train data sent by vehicle-mounted equipment are received.

The train control center receives the information of the track circuit and transmits it to the wireless block center; at the same time, the train control center has the functions of track circuit coding, transponder message storage and calling, safety information transmission between stations, temporary speed limit and so on.

Transponder transmits positioning, inter-stage conversion and other information to vehicle-mounted equipment; at the same time, messages such as line parameters and the temporary speed limit are transmitted to vehicle-mounted equipment to meet the needs of the backup system. The message sent by the transponder has the same meaning as the related content of the information sent wirelessly.

Information provided by ground equipment (such as driving license, line parameters, temporary speed limit, vehicle parameters, etc.) can be extracted and analyzed by the on-board safety computer to generate a dynamic speed curve and monitor the safe operation of trains.

The platform based on trusted data storage architecture can monitor the running state of business system, detect intrusion, extract effective information, and provide a guarantee for the safe and stable operation of the system.

### 5.2. Examples of Water Conservancy Projects

For the storage of credible data in the construction of water conservancy infrastructure, we take the Huangjinxia phase I project of introducing from Han to Wei as an example to carry out the pilot. The Figure 5 shows the mixing building site of the Huangjinxia phase I project. This couplet in Figure 5 illustrates the importance of building the Huangjinxia Dam and the project of diverting Han to Wei. Our scheme is introduced in detail below.

The business process analysis module processes the collected data with java language and converts the data into xml files with unified format through the BML system. Figure 6 shows the xml file with uniform format to be output after the process of business process analysis. Some valid data are marked in the red box, such as data format, event name, form name, organization and so on, and then stored in the blockchain.

The data storage part of the Huangjinxia concrete mixing station is based on blockchain technology, with the hash function, Merkel tree [39] and chain structure as theoretical basis and technical support. In the experiment, SHA 256 hash function [40] is used to electronically identify the data, which simplifies the data scale and ensures the efficiency and accuracy of the data in the process of transmission, location and retrieval.

The operations involved in SHA256 hash function are all logical bit operations, including the following logical functions:(6)$$Ch\left(x,y,z\right)=\left(x\wedge y\right)⊕\left(\neg x\wedge z\right)$$
(7)$$Ma\left(x,y,z\right)=\left(x\wedge y\right)⊕\left(x\wedge z\right)⊕\left(y\wedge z\right)$$

Merkel tree is mainly used for the data storage process of a user system. The two leaf nodes on a component branch represent two complete forms and the forms appear in pairs according to the time sequence of their generation. Because of the structural advantages of Merkel tree, it can compare data efficiently, locate modified data quickly and verify it quickly. The Merkel tree structure designed in this experiment is shown in Figure 7.

Figure 8 shows the tree structure formed after data storage in the blockchain. From top to bottom, the upper chain of the data form is the warehouse surface block. The unit data block includes several stages of a specific production process. A partial data block is a number of attributes included in a unit data block or data packet divided by participating companies. Unit data blocks are other media with archiving requirements, such as files, tables and images, which are included in some data blocks. The generated block information is stored in the form of chain structure, and each block contains the relevant form data summary, timestamp and hash pointer of the previous block to ensure its traceability and non-tampering.

Figure 9a shows the time-consuming situation of data hash encryption. It can be seen from Figure 9a that with the increase of data size, the time-consuming situation also increases. After hash encryption, form hash and warehouse face hash are generated, which are uploaded to the trusted blockchain. Figure 9b shows the time required for storing the hash value in the blockchain. In this system, the total amount of data in the chain of each warehouse is stable at 51 hash records, which takes 200–500 s. The average chain speed of a single hash value is about 4 to 10 s. This speed is mainly limited by the current platform port, and there is still much room for improvement in the future. In addition, the cost of storing a single hash in the blockchain is less than a penny, and the total cost of 51 pieces of data is less than that of CNY 0.1, which is insignificant compared with the benefits brought by data security.

The data check includes an automatic check and a manual check to meet the needs of users. The check compares the newly calculated form hash and warehouse hash with the linked form hash and warehouse hash, and returns the check result. The checking rule is to check the hash value of the library surface first, and it is not necessary to check whether the library surface belongs to the form hash if it is safe. If there is a problem with the surface hash of the warehouse, its branch hash is checked in turn. It can clearly be seen from Figure 9c that under different form hash numbers, the retrieval and inspection time is basically the same, with little change.

In addition, the uplink efficiency of different platforms is also tested, mainly the public chain of Ethereum and the Fabric alliance chain. The results in Figure 9d show that the uplink time consumption of the alliance chain is about 20~30% of that of the common chain under the same data amount, and the uplink efficiency of the alliance chain is higher.

## 6. Conclusions

This paper proposes a trusted storage architecture for national infrastructure data, which ensures the security, integrity, non-repudiation and traceability of national infrastructure data. The architecture has the following advantages: Blockchain technology combined with federal learning technology ensures multiple data backups and fully protects private data security. It can trace data with high speed and efficiency. Through knowledge extraction, useful information can be extracted, and the role of data can be fully exerted. In the experiment part, the qualitative analysis of this architecture and traditional architecture proves that this architecture has certain advantages. In addition, two application examples of a water conservancy project and train control system are listed, which proves the universality and flexibility of this architecture.

## Figures and Tables

**Figure 1 sensors-22-02318-f001:**
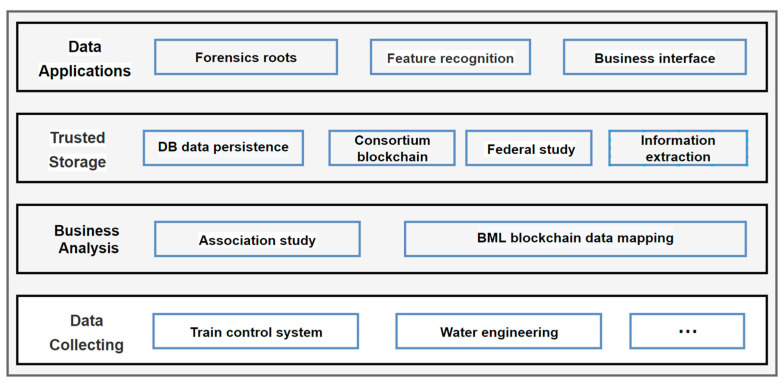
System architecture diagram.

**Figure 2 sensors-22-02318-f002:**
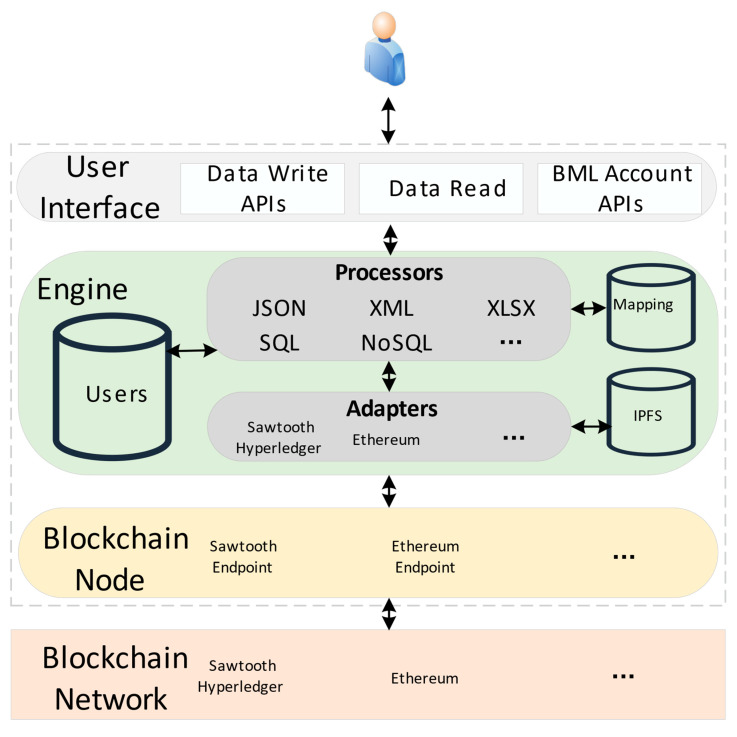
Architecture of BML system.

**Figure 3 sensors-22-02318-f003:**
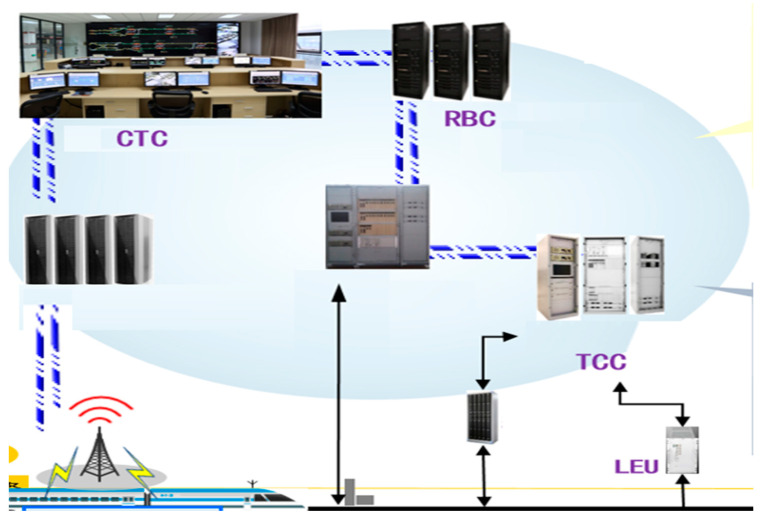
CTCS 3 architecture diagram.

**Figure 4 sensors-22-02318-f004:**
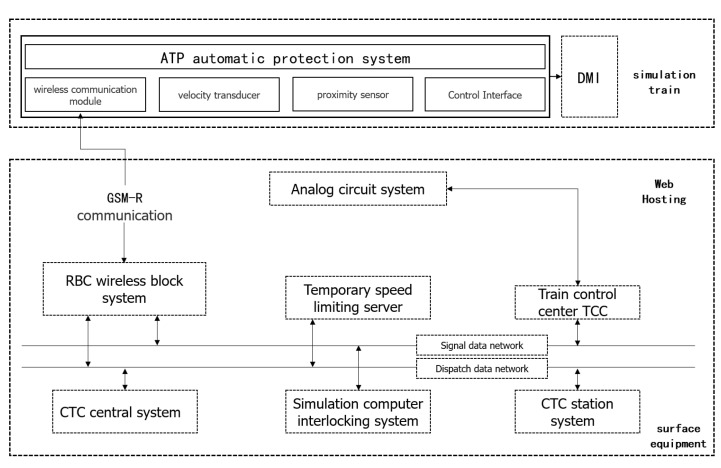
Traffic flow analysis of train control system.

**Figure 5 sensors-22-02318-f005:**
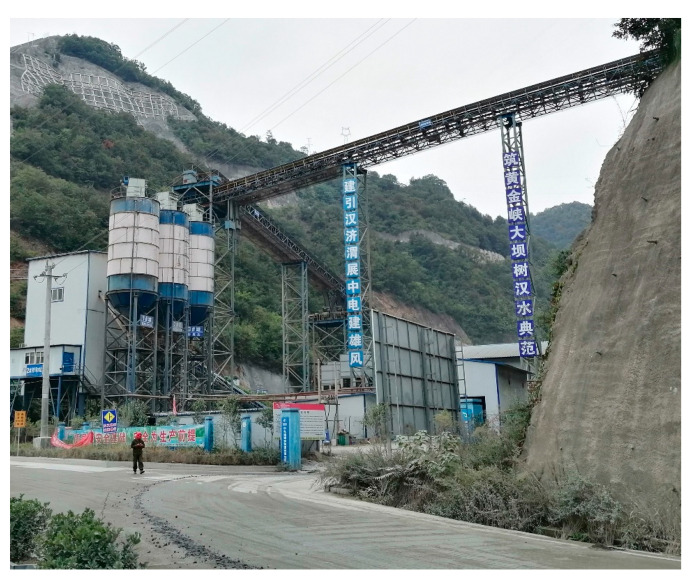
Huangjinxia mixing plant.

**Figure 6 sensors-22-02318-f006:**
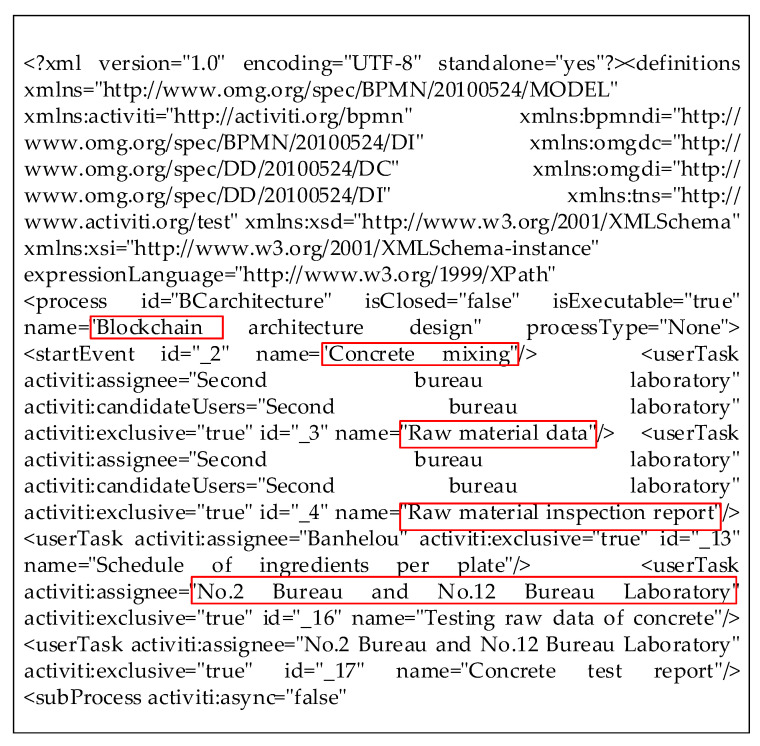
Business xml format.

**Figure 7 sensors-22-02318-f007:**
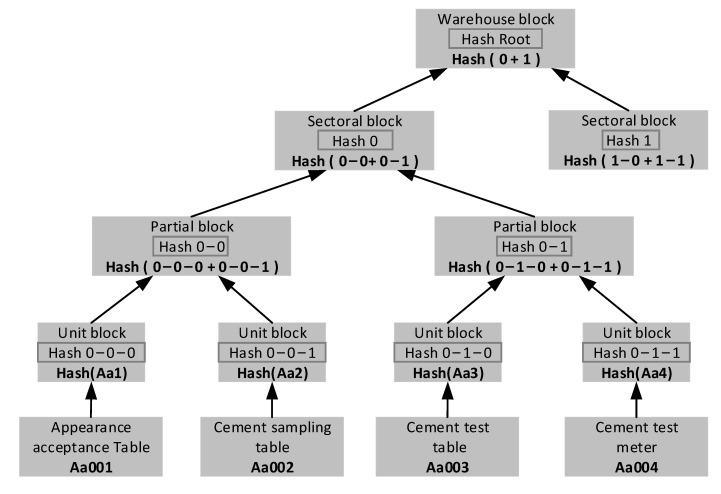
Merkel tree structure diagram.

**Figure 8 sensors-22-02318-f008:**
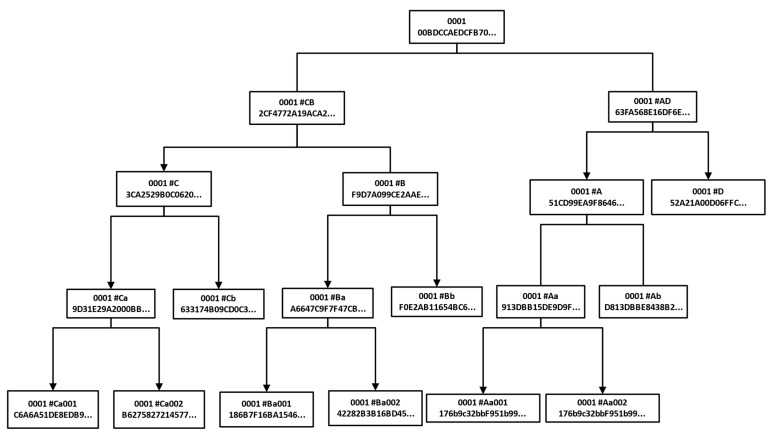
Merkel tree structure diagram of stored data.

**Figure 9 sensors-22-02318-f009:**
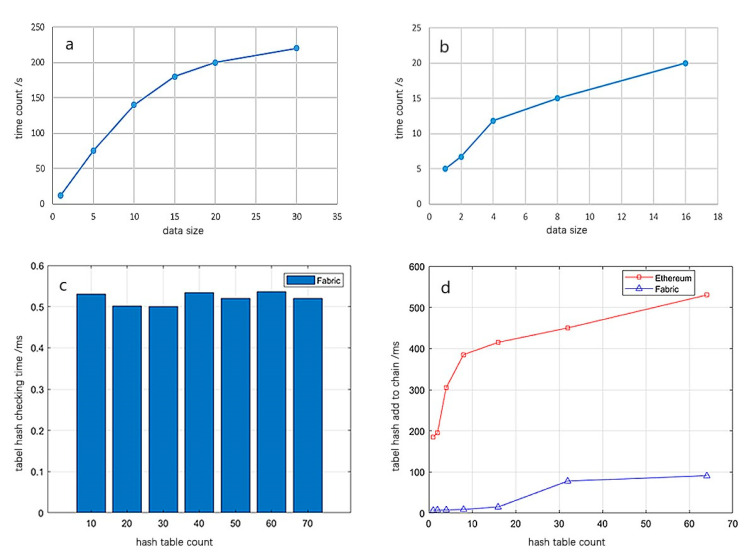
Time consumption of each link of the system and efficiency comparison of different types of chains. (**a**) Data encryption; (**b**) Add hash to chain; (**c**) Data checking; (**d**) The efficiency of contrast.

**Table 1 sensors-22-02318-t001:** Related literature comparison table.

Reference	Main Technology	Main Objective	Application Area
Chen [6]	Blockchain	Data security	Finance
Manimaran et al. [7]	Blockchain	Data security	Finance
Bai [8]	Blockchain and edge computing	Data and computing security	The Internet of Things
Wang et al. [12]	Blockchain	Data security	Vehicle network
Shahbazi [13]	Blockchain and data analysis	Prevent false transactions	Taxi demand service
Martinez et al. [27]	Blockchain and federal learning	Security and sharing of data	-
Lu et al. [28]	Blockchain and federal learning	Data security	Vehicle network
Yin et al. [29]	Blockchain and federal learning	Data security and data cooperation	The Internet of Things

**Table 2 sensors-22-02318-t002:** Experimental configuration table.

Environment	Version	Remarks
Sensor	-	Pressure sensor, displacement sensor, photosensitive sensor, temperature sensor, etc.
Operating system	Ubuntu 20.04	-
Blockchain network	Hyperledger Fabric 2.23	Open source blockchain architecture
Go language	Go 1.14.6	Smart contract development language
Docker	20.10.7	Application container engine
Docker-Compose	1.25.0	Docker tool
Peer	2.2.3	Peer node
Python	3.6	-
CUDA	cuda_10.1.105_418.39_linux	Parallel computing architecture
cuDNN	cudnn-10.1-linux-x64-v7.6.4.38	GPU acceleration library
Pytorch	1.4.0	Deep learning framework
Pysyft	0.2.4	Universal framework of privacy protection deep learning

**Table 3 sensors-22-02318-t003:** A comparison between traditional and trusted storage architectures.

	Data Integrity	Data Provenance	Distributed Storage
Traditional storage methods	√	×	×
Trusted storage methods	√	√	√

## Data Availability

Data can be made available upon request from the corresponding author.

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
