# Peer review of "Trusted Data Storage Architecture for National Infrastructure"

_sensors, 2022, doi:10.3390/s22062318_

Round 1

Reviewer 1 Report

  1. The article is well written and easy to understand in terms of understanding. However requires full proof reading. Formatting is poor. 
  2. Please only add those paper that are from good journal or conferences. Review references because many of them are not standard. 
  3. Novelty of the paper is not clear. Introduction may be improved, adding the highlights and the problem statements. You could discuss the relationship between your solution and past literature. You can add following citations: https://link.springer.com/article/10.1007/s12652-021-03459-4, https://www.mdpi.com/1424-8220/22/3/832
  4. I suggest add a comparative table in ''Related Literature'' to contrast your solution in front of related works.
  5. The conclusion needs improvements towards major claimed contribution. 
  6. The quality of the figures and tables should be improved.
  7. Provide the experimental setup and the tools used for the study. If possible provide a simulation parameters table.

Reviewer 2 Report

This article offers a novel blockchain-based trusted storage architecture. According to the business features of infrastructure building, the trusted storage architecture for national infrastructure data is primarily separated into data collection, business flow analysis, trusted storage, and data application modules. Blockchain technology is a distributed network data management system that uses cryptography and a distributed consensus mechanism to assure network transmission and access security, multi-party data maintenance, cross-verification, and overall network consistency that is difficult to tamper with. Typical databases are employed for data durability, while alliance chains are used for knowledge extraction. The application layer can then do forensic tracing, feature recognition, and other traditional business interface functions.

This article has a medium length and a good structure. The article is not so well written (for example, on page 3, I've noticed these: "To sum up, in this paper, We have made the following two contributions We have made the following two contributions" and "Independent and credible."). I'd suggest modifying these two text parts. All the figures are intelligible, but the "User Interface" text (from Figure 2) should be white, Figure 4 should be extended, Figure 6 is missing some boxes lines, and the font of the text from those boxes should've been more prominent. Table 1 seems to use an unnecessary big font. Figures numbering is wrong (Figure 7, 8 and then 7 appears again, and then eight appears AGAIN). Figure 8 from page 14 has its title in bold, but the rest does not. Shouldn't all use an identical font? Here "presented by the proposed BML mapping language is as shown in Figure3", shouldn't it be Figure 4? Some figures (but Figures 1, 2, 3, 1st and 2nd 8) should be lowered with ~1 row because they get into the text above them.

The article should've been better verified before publishing, but it has an accurate structure. The abstract terms are adequately used. The title is concise, but it does not give the reader the paper's keywords so that the person who reads it doesn't have a clear idea of the paper's content from the beginning. The Abstract and the Introduction chapters include enough details about the article's subject matter. In the Conclusion section, I've noticed the following text structure: "In this work, ..,  in the first place. Secondly, according to .. operations,  Secondly, it has been ..  infrastructure." which tells that the paragraph should be rewritten/rephrased. The pseudocode description from the bottom of page 9 doesn't look good; I think it should appear as text that follows the paper's template.

Definite and indefinite articles are missing (or added where they’re not needed). The authors should pay more attention to the grammar rules and the spelling check: “flow analysis in the first place, Secondly, ” - flow analysis in the first place. Secondly, “on the one hand, traditional database is used for” - on the one hand, a traditional database is used for “on the other hand, alliance chain is used for” - on the other hand, an alliance chain is used for “can monitor the running state of business system” - can monitor the running state of the business system “to ensure the driving safety” - to ensure driving safety “as theoretical basis” - as a theoretical basis To be honest, some phrases seem too long and hard to understand or keep up with their content (e.g., “In this work, we proposed a trusted storage architecture, geared to the needs of national infrastructure data according to the characteristics of the main business of infrastructure construction, is divided into data acquisition, analysis of the business flow, trusted storage, and data application module, the underlying data collected by sensors, paper files, such as audio and video after the business flow analysis in the first place”; “Secondly, according to the characteristics of business flow, on the one hand, a traditional database is used for data persistence, on the other hand, an alliance chain is used for knowledge extraction, and then the application layer can carry out forensic tracing, feature recognition and other traditional business interface operations.“). I’d shorten such extended phrases.

The references should include related work regarding time critical IoT, for example:

- Ghosh, Shreya, et al. "Mobi-iost: mobility-aware cloud-fog-edge-iot collaborative framework for time-critical applications." IEEE Transactions on Network Science and Engineering 7.4 (2019): 2271-2285.

- Štefanič, Polona, et al. "SWITCH workbench: A novel approach for the development and deployment of time-critical microservice-based cloud-native applications." Future Generation Computer Systems 99 (2019): 197-212.

Reviewer 3 Report

The paper looks like a quick shot: mismatches in numbering the figures; missing dots at the end of sentences; symbols, used in formulas, are not explaint; typos and so on.
Please read the paper carefully before resending it for a second review.

Here are some examples (Line numbers in brackets):
-Figures: (49), (427) figure number is missing,   (497) and (554) same number: figure 7, also figure 8 (525) and (573),

How figure 1 (205) corresponds to figure 7 (554)

-long sentences: (207-210), (257-261), (640-647)

- missing dots between sentences: (270), (361), (426), (529)

- formula without numbers, e.g. (311), symbols used  are not explaint (X,Y,J;D)

what are the symbols in formula (327) and (332) fi w∈Rd ,Fk?

(329) what is the difference between "kth participant" and "costumer K"? If none, please don't use different symbols (k, K).

(361) x ∩ y = ?

(471) Merkel tree:  errors in branch-block Hash 0-0 and 0-1:
Hash (0-0-0+0-0-1) not Hash (0-0+0-0-1)
Hash (0-1-0+0-1-1) not Hash (0-1-0+1-1)
as well as in unit block Hash :
Hash (1-0 + 1-1) not (1-0+0-1)

closing bracket without opening (370)

Round 2

Reviewer 1 Report

Authors updated the paper as per my previous comments and no further update requires from my side.

Author Response

Thank the reviewers for their comments on our manuscript entitled "Trusted Data Storage Architecture of National Infrastructure" (ID: sensors-1600417). These comments are valuable, which are very helpful to revise and improve our papers, as well as the important guiding significance to our research.

We tried our best to improve the manuscript and made some changes to it. Improve the introduction, research design, method description, result analysis, conclusion and other parts. And checked spelling and grammar. These changes will not affect the content and framework of the paper. We sincerely thank the reviewers for their enthusiastic work and hope that the amendments can be approved. Thank you again for your comments and suggestions.

Reviewer 3 Report

Most of the remarks are now sufficient corrected, but still the sentence (item 7 in the last review) "In a horizontal federal learning system, let ?? represent the data set owned by the kth participant, and ?? represent the index set of data points located in customer K." operates with two versions of letter K and k.

Again, the questions: is there a difference between k-th participant and customer K? is there a difference between K and k?

If this question is suffuciently answered, the paper can be published.
